# Population Pharmacokinetic Model of Vitamin D_3_ and Metabolites in Chronic Kidney Disease Patients with Vitamin D Insufficiency and Deficiency

**DOI:** 10.3390/ijms252212279

**Published:** 2024-11-15

**Authors:** Stacey M. Tuey, Avisek Ghimire, Serge Guzy, Linda Prebehalla, Amandla-Atilano Roque, Gavriel Roda, Raymond E. West, Michel B. Chonchol, Nirav Shah, Thomas D. Nolin, Melanie S. Joy

**Affiliations:** 1Department of Pharmaceutical Sciences, Skaggs School of Pharmacy and Pharmaceutical Sciences, University of Colorado, Aurora, CO 80045, USAavisek.ghimire@cuanschutz.edu (A.G.); amandla.atilanoroque@uchealth.org (A.-A.R.); gavriel.roda@cuanschutz.edu (G.R.); 2Pop—Pharm Pharmacometrics Service, Albany, CA 94706, USA; poppharm2@gmail.com; 3Center for Clinical Pharmaceutical Sciences, Department of Pharmacy and Therapeutics, School of Pharmacy, University of Pittsburgh, Pittsburgh, PA 15261, USA; lprebeh@pitt.edu (L.P.); nolin@pitt.edu (T.D.N.); 4Division of Renal Diseases and Hypertension, University of Colorado, Aurora, CO 80045, USA; michel.chonchol@cuanschutz.edu; 5Department of Medicine Renal Electrolyte Division, University of Pittsburgh, Pittsburgh, PA 15261, USA; nas65@pitt.edu

**Keywords:** chronic kidney disease, cholecalciferol, vitamin D deficiency, population pharmacokinetic model

## Abstract

Vitamin D insufficiency and deficiency are highly prevalent in patients with chronic kidney disease (CKD), and their pharmacokinetics are not well described. The primary study objective was to develop a population pharmacokinetic model of oral cholecalciferol (VitD_3_) and its three major metabolites, 25-hydroxyvitamin D_3_ (25D_3_), 1,25-dihydroxyvitamin D_3_ (1,25D_3_), and 24,25-dihydroxyvitamin D_3_ (24,25D_3_), in CKD patients with vitamin D insufficiency and deficiency. CKD subjects (*n* = 29) were administered one dose of oral VitD_3_ (5000 I.U.), and nonlinear mixed effects modeling was used to describe the pharmacokinetics of VitD_3_ and its metabolites. The simultaneous fit of a two-compartment model for VitD_3_ and a one-compartment model for each metabolite represented the observed data. A proportional error model explained the residual variability for each compound. No assessed covariate significantly affected the pharmacokinetics of VitD_3_ and metabolites. Visual predictive plots demonstrated the adequate fit of the pharmacokinetic data of VitD_3_ and metabolites. This is the first reported population pharmacokinetic modeling of VitD_3_ and metabolites and has the potential to inform targeted dose individualization strategies for therapy in the CKD population. Based on the simulation, doses of 600 International Unit (I.U.)/day to 1000 I.U./day for 6 months are recommended to obtain the target 25D_3_ concentration of between 30 and 60 ng/mL. These simulation findings could potentially contribute to the development of personalized dosage regimens for vitamin D treatment in patients with CKD.

## 1. Introduction

Vitamin D_3_ (VitD_3_) is a fat-soluble prohormone essential for maintaining calcium and phosphorus homeostasis and overall bone health [1]. The majority of VitD_3_ is produced in the body from 7-dehydrocholesterol upon skin exposure to ultraviolet B light from the sun. In addition, vitamin D can be obtained through diet or supplements in the form of ergocalciferol (VitD_2_) or cholecalciferol (VitD_3_). Regardless of its source, once in circulation, VitD_3_ is transported by vitamin D-binding protein (DBP) to the liver where it is converted by the cytochrome P450 (CYP) enzyme CYP2R1 to form 25-hydroxyvitamin D_3_ (25D_3_), the major circulating form of VitD_3_. The uptake of 25D_3_ bound to DBP into the proximal tubule of the kidney occurs via megalin-mediated endocytosis [2]. Once inside the kidney, 25D_3_ undergoes a second hydroxylation step by CYP27B1 to form the active metabolite 1,25-dihydroxyvitamin D (1,25D_3_), or it is catabolized to 24,25-dihydroxyvitamin D (24,25D_3_) by CYP24A1 [1]. Minor metabolites of VitD_3_ have also been reported [3]. The binding of active 1,25D_3_ to the vitamin D receptor (VDR) in the cytoplasm leads to the heterodimerization of VDR to the retinoic acid X receptor [4]. This complex translocates to the nucleus and binds to the vitamin D response element (VDRE) on the promoter region of target DNA sequences and regulates over 200 genes responsible for a wide range of biological actions that include cell proliferation [5], renin production [6], drug metabolism [7,8,9,10], and apoptosis [11,12].

VitD_3_ deficiency is well recognized as a worldwide public health problem [13]. Patients with chronic kidney disease (CKD) are among the most vulnerable populations at risk for VitD_3_ deficiency with prevalence rates of up to 80% previously reported [14,15,16,17]. Evidence in CKD patients has reported that low levels of VitD_3_ are associated with secondary hyperparathyroidism (SHPT), mineral and bone disorders, cardiovascular risks, and all-cause mortality [18,19,20]. Experts have defined VitD_3_ insufficiency in the general population as serum 25D_3_ levels between 20 and <30 ng/mL and deficiency as <20 ng/mL [21,22]. Generally, treatments seek to target 25D_3_ levels of >30 ng/mL for the general population [23]. However, there remains a lack of consensus regarding target 25D levels and optimal replacement and maintenance dosing strategies in the CKD population. Current guidelines for treating VitD_3_ insufficiency and deficiency recommend that patients with non-dialysis-dependent CKD stages 1–5 should follow the same VitD_3_ dosing strategies recommended for the general population [24,25]. As such, the Kidney Disease Outcome Quality Initiative (KDOQI) suggests oral cholecalciferol 1000–2000 international units (I.U.)/day for VitD_3_ repletion, but it also acknowledges that CKD patients may require more aggressive dosing regimens [24].

The CKD population exhibits substantial variation in renal function, body composition, comorbidities, and concomitant medications that complicate dose–response relationships for VitD_3_. There is currently a lack of studies relating VitD_3_ levels to clinical outcomes, which has made it difficult to formulate precise guidelines for VitD_3_ dosing regimens and repletion targets in patients with CKD. Robust and comprehensive population pharmacokinetic models to characterize the disposition of VitD_3_ and its metabolites remain scarce in the general and CKD populations [26,27,28,29,30]. Importantly, a population pharmacokinetic assessment of VitD_3_ and multiple metabolites in CKD patients has not been performed. The development of a population pharmacokinetic model for VitD_3_ and its major metabolites may permit the identification of individual factors (e.g., covariates) affecting pharmacokinetic parameters and provide a rationale for the enhanced precision of VitD_3_ dosing. The aim of the current study was to develop a population pharmacokinetic model of VitD_3_ and three major metabolites in CKD subjects with total 25D_3_ levels below 30 ng/mL after receiving a single 5000 I.U. oral dose of cholecalciferol. 

## 2. Results

### 2.1. Study Participants

A total of 29 patients with CKD and VitD_3_ deficiency (25D_3_ < 30 ng/mL) were included in this study. The baseline characteristics of the 29 participants are presented in Table 1. Of these patients, 59% were female, the median age (range) was 61 (29–73) years, and the median weight (range) was 92.0 (70.7–135.3) kg. The median (range) estimated glomerular filtration rate (eGFR) was 37 (11–97) ml/min/1.73 m^2^, the median body mass index (BMI) was 32.6 (25.6–43.4) kg/m^2^, and the median (range) baseline total 25D_3_ was 18 (7–29) ng/mL. Several targeted patient parameters were assessed given their potential impact on VitD_3_ metabolism and concentrations. Weight and BMI can have an inverse relationship to VitD_3_ concentrations, as tissue distribution increases with increased body fat. Age-related physiological changes and the eGFR, a marker of renal function, can impact the metabolism of VitD_3_. A total of 310 plasma VitD_3_ and metabolite concentrations were included in the analysis for model development. Of these, 212 observations for the parent VitD_3_ were below the limit of quantification (BLQ).

### 2.2. Base Model

The final pharmacokinetic model for the parent, VitD_3_, and metabolites, 25D_3_, 1,25D_3_, and 24,25D_3_, is depicted in Figure 1**.** The models were executed with the Quasi-Random Parametric Expectation Maximization (QRPEM) engine in Phoenix^®^ NLME. Based on the objective function value (OFV), the M3 method was selected for handling BLQ data. 

After the base model of the parent compound was established, the model was extended to the three major metabolites. As the parent VitD_3_ was administered alone without the administration of metabolites, the fraction of metabolite formation and the volume of metabolites were not identifiable. Therefore, the fraction on VitD_3_ converted to 25D_3_ (fm_1_) was fixed to 1, and the model assumed no alternative elimination pathways for VitD_3_. The 25D_3_ data were best explained by a one-compartment model with first-order formation from VitD_3_. The parameters for 25D_3_ were the volume of distribution (V_m1_), baseline 25D_3_ concentration (C0_m1_), and clearance (CL_m1_). The fraction of 25D_3_ converted to 1,25D_3_ (f_m2_) was fixed to 0.017, and the remaining 25D_3_ was assumed to be eliminated through conversion to 24,25D_3_. 1,25D_3_ and 24,25D_3_ were both best described by a one-compartment model for first-order formation from 25D_3_ and first-order elimination. The parameters for the parent were ka, k_endog_, apparent central volume of distribution (Vc/F_VitD3_), peripheral volume of distribution, (Vp/F_VitD3_), intercompartmental clearance (Q_VitD3_), baseline VitD_3_ concentration (C0), and apparent clearance (CL/F_VitD3_). IIV terms for C0, Vc/F_VitD3_, and CL/F_VitD3_ were included. The parameters for 1,25D_3_ and 24,25D_3_ were the volume of distribution (V_m2_ and V_m3_, respectively), baseline 1,25D_3_ and 24,25D_3_ concentration (C0_m2_ and C0_m3_, respectively), and clearance (CL_m2_ and CL_m3_, respectively) (Table 2).

A proportional error model best explained the residual variability for the parent and major metabolites (Table 2). Due to the complexity of the model, fixed values for ka, k_endog_, Vp/F_VitD3_, and Q/F_VitD3_ were employed. The selected parameters ka (0.323 h^−1^), Vp/F_VitD3_ (2333 L), and Q/F_VitD3_ (0.185 L/h) were fixed to value from the literature [29] or from previous iterations of the model. Fixing parameter estimates from previous iterations of the model increased the precision of primary pharmacokinetic parameter estimates and did not affect the OFV. The pharmacokinetic parameters determined for VitD_3_ were as follows: baseline concentration (C0) of 2.54 ng/mL (0.98 nmol/L) with CV 41.7%; VitD_3_ apparent central volume of distribution (Vc/F_VitD3_) of 21.3 L, with CV 22.2%; and VitD_3_ apparent clearance (CL/F_VitD3_) of 1.4 L/h with CV 42.4%. For 25D_3_, the baseline concentration (C0_m1_) was 108.57 ng/mL (43.5 nmol/L) with CV 4.15%; volume of distribution (V_m1_) was 58.3L; and clearance (CL_m1_) was 0.02 L/h. The baseline concentration of 1,25D_3_ (C0_m2_) was 0.48 ng/mL (0.20 nmol/l) with CV 6.9%; volume of distribution (V_m2_) was 71.5 L with CV 206.8%; and clearance (CL_m2_) was 0.08 L/h. For 24,25D_3_, the baseline concentration (C0_m3_) was 528 ng/mL (2.2 nmol/L) with CV 9.4%; volume of distribution (V_m3_) was 105.2 L with CV 140.5%; and clearance (CL_m3_) was 0.4 L/h with CV 53.4%. Overall, the population pharmacokinetic parameters were estimated with adequate precision with the exception of V_m2_ and V_m3_. There was substantial unexplained variability for 25D_3_ plasma levels, as illustrated by the estimated proportional residual error of 65.7%. For VitD_3_, 1,25D_3_, and 24,25D_3_, the proportional residual errors were <20%.

### 2.3. Covariate Model

Based on the visual inspection of covariate plots, several were worthy of further investigation, including the effect of weight on the baseline concentration of 1,25D_3_ (C0_m2_), BMI on the volume of 1,25D_3_, and the eGFR on the baseline concentration of 1,25D_3_ (C0_m2_) and 24,25D_3_ (C0_m3_). The effects of these identified covariates were evaluated for their effects on the model by forward addition and backward elimination. No covariates evaluated led to significant influences on parent and metabolite pharmacokinetics in terms of the statistical significance criteria as specified in the Methods section (4.7).

### 2.4. Model Evaluation

Goodness-Of-Fit (GOF) plots for VitD_3_, 25D_3_, 1,25D_3_, and 24,25D_3_ are depicted in Figure 2. These figures show adequate agreement between the observed, individual predicted, and population predicted VitD_3_, 25D_3_ 1,25D_3_, and 24,25D_3_ concentrations (Figure 2A,B). The conditional weighted residual vs. predicted concentrations and time did not show any specific patterns for the parent or metabolites (Figure 2C,D). The weighted residuals were close to y = 0, and most of the values were between y = −2 and 2.

Visual Predictive Checks (VPCs) were generated for VitD_3_ and metabolites using 200 replicates and are presented in Figure 3A–D.The 5th, 50th, and 95th percentiles of the simulated data obtained from the VPCs were plotted against observed concentrations. In general, the 5th, 50th, and 95th percentiles of the observed concentrations were in agreement with the predicted concentration percentiles, demonstrating that the pharmacokinetics of VitD_3_ and its metabolites were adequately described by the final model. However, 24,25D_3_ was underpredicted, particularly at higher concentrations, which may be due to sparse data in that region (Figure 3A–D).

### 2.5. Simulations 

The final population pharmacokinetic model was used to simulate the 25D_3_ concentrations using clinically applicable cholecalciferol dosing regimens of 600 I.U./day, 1000 I.U./day, 2000 I.U./day, 5000 I.U./day, and 10,000 I.U./day for 6 months. The simulations were conducted based on cholecalciferol dosage regimens used in the clinic and represented a range of low, middle, and high doses. As most of the observed VitD_3_ data were BLQ, and 25D_3_ being the major circulating metabolite of VitD_3_, we focused on a full range of simulated concentrations for 25D_3_ (Figure 4). The mean 25D_3_ concentration over time for each dose level was plotted (Figure 4A–E).

Based on the simulation results and the plot, the mean concentration of 30 ng/mL was reached within 1488 h (62 days), 840 h (35 days), 360 h (15 days), 144 h (6 days), and 96 h (4 days) for doses of 600 I.U./day, 1000 I.U./day, 2000 I.U./day, 5000 I.U./day, and 10,000 I.U./day, respectively. The maximum 25D_3_ concentration (mean ± s.d.) by the end of the treatment was 38.1 ± 3.5 ng/mL, 54.1 ± 3.2 ng/mL, 95.3 ± 4.3 ng/mL, 218.5 ± 7.1 ng/mL, and 424.03 ± 10.4 ng/mL for doses of 600 I.U./day, 1000 I.U./day, 2000 I.U./day, 5000 I.U./day, and 10,000 I.U./day, respectively. 

## 3. Discussion

As the prevalence rates of CKD continue to rise [31], there is high priority to improve the treatment strategies of symptoms and associated complications. Individuals with CKD are at greater risk for VitD_3_ deficiency compared to the general population, and an altered metabolism of VitD_3_ has been proposed to be a contributing factor [32,33,34,35]. There is considerable debate regarding target VitD_3_ levels and strategies for VitD_3_ supplementation in CKD. The 2003 KDOQI guidelines recommend 25D_3_ concentrations ≥ 30 ng/mL in CKD stages 3 and 4 to prevent SHPT [36]. A report from a Scientific Workshop sponsored by the National Kidney Foundation (NKF) suggested that 25D_3_ adequacy should be classified as concentrations >20 ng/mL without evidence of counter-regulatory hormone activity such as elevated parathyroid hormone (PTH) levels [37]. In addition, 25D_3_ concentrations <15 ng/mL should be treated regardless of PTH level, and patients with 25D_3_ between 15 and 20 ng/mL may not need VitD_3_ treatment if counter-regulatory hormone activity is not observed [37]. However, a more recent cross-sectional analysis in stages 1–5 CKD patients (*n* = 14,289) found that 25D_3_ levels of 42–48 ng/mL were actually necessary to lower PTH levels [38]. This study also reported that higher target concentrations of 25D_3_ were not associated with an additional risk of hypercalcemia and hyperphosphatemia. Guidelines suggest that patients with CKD stages 1–5 and VitD_3_ insufficiency or deficiency should follow the same supplementation strategies recommended for the general public. The KDOQI recommends 1000–2000 I.U./day of VitD_3_ but acknowledges that CKD patients may require a more aggressive treatment plan [24]. The Endocrine Society recommends VitD_3_ 1500–2000 I.U./day for adults and three times the recommended dose for individuals with a BMI > 30 kg/m^2^ [39]. However, a retrospective cohort study in stages 2–5 CKD subjects (*n* = 309) reported that after treatment with 10,000–50,000 I.U./week of VitD_3_, 42.7% of patients failed to attain increased 25D_3_ levels above 40 ng/mL [40]. Taken together, it is evident that strategies to inform the dose–concentration relationships of VitD_3_ in the CKD population are needed. Approaches using population pharmacokinetic models have the potential to predict the plasma concentrations of VitD_3_ metabolites following VitD_3_ dosage regimens. Given the sparsity of information on the pharmacokinetics of VitD_3_, this study focused on developing a population pharmacokinetic model for VitD_3_ and its major metabolites, 25D_3_, 1,25D_3_, and 24,25D_3_, in CKD subjects with VitD_3_ deficiency following the administration of a single oral 5000 I.U. dose of VitD_3_. To our knowledge, this is the first study to simultaneously model the pharmacokinetics of VitD_3_ and its major metabolites, 25D_3_, 1,25D_3_, and 24,25D_3_, using a nonlinear mixed effects population modeling approach.

In the current study, a population pharmacokinetic approach described VitD_3_ and its major metabolites and filled a gap in the current understanding regarding VitD_3_ pharmacokinetics in CKD. This approach provides a framework for investigating the relationships between dosing strategies and the attainment of targeted concentrations to improve outcomes in this population. A two-compartment model for VitD_3_ and a one-compartment model for each metabolite were used. The model fits the patient data well, as demonstrated by VPC graphs. Given the substantial proportion of BLQ data for VitD_3_, we examined how the two BLQ data treatment approaches affected the model estimates. Based on the OFV, the M3 method to treat BLQ data was incorporated into the model.VitD_3_ entered into the central compartment through the oral absorption of the administered dose (ka) and constant endogenous production (k_endog_), the latter of which is a function of the average baseline concentration and the elimination of VitD_3_. While the model used a fixed endogenous rate of 0.55 nmol/h, in reality, endogenous production can fluctuate due to many factors including season and lifestyle [41,42]. However, given that all participants in this study had low levels of VitD_3_, endogenous production is likely a minor contribution to overall VitD_3_ concentrations. ka was fixed to 0.054 h^−1^ based on estimations derived from previous iterations of the model as there were inadequate data in the absorption phase for its estimation. To reduce the complexity of the model, we chose to focus on the estimation of the primary pharmacokinetic parameters, the apparent central volume of distribution and apparent central clearance. Therefore, the intercompartmental clearance of VitD_3_ was fixed to 0.44 L/h. The fixed values selected for this parameter were based on estimates from iterations of the model which resulted in better model performance compared to using fixed values reported from the scarce literature [29]. The apparent central volume of distribution of VitD_3_, Vc/F_VitD3_, estimate was 21.3 L. VitD_3_ is a lipophilic compound with a reported estimated partition coefficient (log P) of 8.8 [43], and adipose tissue is the major storage site of VitD_3_ and its metabolites [44,45]. Ocampo-Pelland et al. developed a population pharmacokinetic model for VitD_3_ and the 25D_3_ metabolite using a model-based meta-analysis of data from 57 studies representing 5395 healthy or osteoporotic adult subjects [29]. They reported a two-compartment model for VitD_3_, and the estimate for the central volume of distribution was 15.6 L, which is in agreement with the estimate in our model. The estimated apparent oral clearance of VitD_3_, CL/F_VitD3_, in our model was 1.42 L/h. Ocampo-Pelland et al. reported a nonlinear, Michaelis–Menten elimination of VitD_3_ based on data from 57 studies in which subjects received multiple doses of VitD_3_ ranging from 400 to 300,000 I.U./day for a minimum of 4 weeks. They reported that the VitD_3_ maximum rate of elimination was 1.62 nmol/h, and the Michaelis–Menten constant was 16.6 ng/mL (6.4 nmol/L) [29]. Nonlinear elimination was not observed in the current study where participants received a single 5000 I.U. dose of VitD_3_, suggesting that concentrations were below the level of saturation.

In the current model, the pharmacokinetics of 25D_3_ (the first major metabolite in the pathway) was described by a one-compartment model with first-order formation and first-order clearance. The volume of distribution of 25D_3_, V_m1_, was estimated to be 58.3 L, suggesting distribution into tissue. A previous pharmacokinetic model for 25D_3_ from patients (*n* = 422) diagnosed with human immunodeficiency virus (HIV) reported a volume of distribution of 178 L [27]. The data in this later study were retrospectively collected from patient records where some patients received a median VitD_3_ dose of 63,302 I.U. per month. Another pharmacokinetic model from renal transplant recipients (*n* = 49) who received 100,000 I.U. VitD_3_ every 2 weeks followed by every 2 months until 1-year post-transplant reported an estimated volume of distribution of 237 L for 25D_3_ [26]. While differences in study populations may contribute towards the discrepancy in the reported 25D_3_ volume of distribution, in these previous models, patients received multiple and higher doses of VitD_3_ than participants in the current study. This may suggest that the distribution of 25D_3_ is dose- or concentration-dependent where at higher concentrations, larger amounts of 25D_3_ are stored in tissue. Nearly identical findings were disclosed in a study that investigated VitD_3_ and 25D_3_ concentration in the abdominal subcutaneous fat tissue of participants who received weekly 20,000 I.U. VitD_3_ vs. placebo for 3–5 years [46]. This study found that the median concentrations of 25D_3_ in fat tissue were 3.8 ng/g in subjects given VitD_3_ vs. 2.5 ng/g in the placebo group. The population estimate of 25D_3_ clearance in the current model was 0.02 L/h. Pharmacokinetic models in young healthy adults [28], HIV patients [27], and renal transplant recipients [26] reported 25D_3_ clearance estimates of 0.01, 0.12, and 0.10 L/h, respectively. The slower clearance for 25D_3_ estimated in the current model could indicate an impaired metabolism of 25D_3_ to 1,25D_3_ or 24,25D_3_ through CYP24A1 or CYP27B1, respectively. Reduced CYP function has been reported in patients with CKD [47,48,49]. Given that the reported half-life of VitD3 is approximately 2 months [50] and 25D_3_ ranges from 2 weeks to 2 months [51,52], the current study likely did not fully capture the elimination phase of 25D_3_ as sampling beyond 14 days was not feasible.

While 1,25D_3_ and 24,25D_3_ are not routinely measured in a clinical setting, abnormal levels have been reported in CKD [33,53,54]. Therefore, the characterization of these metabolites may provide important information for understanding alterations in metabolism pathways secondary to CKD and for optimizing dosing strategies in this population. The mean parameter estimates of the volume of distribution of 1,25D_3_ and 24,25D_3_ were 71.5 L and 105.2 L, respectively. However, there was a large degree of uncertainty in these parameter estimates with CV% >100%. For the metabolite models to be fully identifiable, the fraction of 25D_3_ metabolized to 1,25D_3_ (f_m2_) was assumed to be 0.017, and the fraction of 25D_3_ metabolized to 24,25D_3_ was 1-f_m2_. Since the information of the percent conversion of VitD_3_ to these metabolites was absent in the literature, the fractions used in this model were based on estimates from a published PBPK model that used data from subjects without kidney disease [55]. We are currently unable to ascertain whether the metabolite fractions in the current study are representative of alterations in CKD due to the lack of comparison data. Another approach for identifiability is to use a fixed value for the volume of distribution of metabolites which allows other parameters to be estimated relative to the fixed value [56]. However, given the limited information on the pharmacokinetics of VitD_3_ reported in the literature, particularly for the two dihydroxy metabolites, we preferred using a fixed value for the fraction metabolized over alternative approaches. This parameterization does assume a constant fraction metabolized for each metabolite, which we acknowledge is a limitation of this model given that this fraction is likely to vary based on factors such as the baseline VitD_3_ level and the regulation of CYP enzymes responsible for VitD_3_ metabolism [57]. Therefore, the volume of distribution of the metabolites in the current model should be interpreted with the assumption that the fraction of VitD_3_ metabolized to 25D_3_ and the fraction of 25D_3_ metabolized to 1,25D_3_ and 24,25D_3_ were the same for all participants.

Despite the relatively small number of study patients, several covariates were tested (weight; BMI; age; gender; race; eGFR; genetic polymorphisms in *CYP2R1*, *CYP27B1*, *CYP24A1*, *VDR*, and *GC*; and serum protein levels of PTH and fibroblast growth factor 23 (FGF-23)) to determine their influence on the pharmacokinetics of VitD_3_ and its metabolites. A visual inspection of covariate plots suggested that lower baseline concentrations of 24,25D_3_ were associated with a lower eGFR. A large cross-sectional study (*n* = 9596) reported similar findings; a lower eGFR was strongly associated with reduced VitD_3_ catabolism, leading to lower 24,25D_3_ concentrations [58]. While none of the covariates assessed in the current study significantly affected parameter variability, it is plausible that this study lacked enough statistical power to detect significant covariates, and therefore, we cannot rule out their influence on the disposition of VitD_3_ and its metabolites. Certain covariates such as weight and BMI could have clinical relevance for VitD_3_ pharmacokinetics. Given the large number of parameters in addition to the smaller sample size, more sophisticated approaches may be necessary to determine influential covariates. Numerous published studies have found weight and obesity to be associated with lower serum concentrations of VitD_3_ and 25D_3_ [59,60,61]. Incremental increases in serum VitD_3_ following whole-body ultraviolet radiation were 57% lower in subjects with a BMI ≥ 30 kg/m^2^ vs. subjects with a BMI < 25 kg/m^2^ [59]. The same study reported an inverse correlation of BMI with serum VitD_2_ concentrations following an oral dose of 50,000 I.U. VitD_2_ [59]. The influence of obesity is likely due to increased distribution in the adipose tissue in obese patients which in turn decreases the bioavailability of VitD_3_. While the current study did not find weight or BMI to have a significant association with VitD_3_ or metabolite pharmacokinetic parameters, all subjects in the current study had a BMI of >25 kg/m^2^. Sun exposure, geographical location, diet, and seasonal variation are also possible sources contributing to the high IIV that were not accounted for in this model. The loss of DBP-bound VitD_3_ and metabolites in the urine because of proteinuria was also not assessed in the current model. Hence, the influence of proteinuria on VitD_3_ levels remains unclear, and conflicting results have been reported [62,63,64]. Regardless, high IIV on some pharmacokinetic parameters of VitD_3_ and metabolites in the current model could underline the importance of further investigation into factors associated with variability.

Considering that 25D_3_ levels of >30 ng/mL indicate clinical repletion [11,13,21] and levels of up to 60 ng/mL is exemplary [11,39], a simulation was performed to ascertain the dose regimen of cholecalciferol necessary to achieve the 25D_3_ serum concentration targets of 30 ng/mL and 60 ng/mL. 

The final population pharmacokinetic model was used to run the simulation with the standard dosing regimens of 600 I.U./day, 1000 I.U./day, 2000 I.U./day, 5000 I.U./day, and 10,000 I.U./day for 6 months of treatment. The target concentration of 30 ng/mL was achieved the most quickly with the dose of 10,000 I.U./day and slowest with the dose of 600 I.U./day. However, the maximum concentration of 25D_3_ in the range of 30–100 ng/mL was achieved for 600 I.U/day to 2000 I.U./day, and this concentration range is generally considered safe [23,39]. If a concentration range of between 30 and 60 ng/mL is targeted, the dosing of 600 I.U/day or 1000 I.U/day of VitD_3_ for 6 months would achieve this range to reduce complications associated with VitD_3_ deficiency in the CKD population. 

## 4. Methods

### 4.1. Study Design

Study subjects were admitted for a 12 h stay followed by additional visits at 24, 48, 168, and 336 h. All visits were paid to the Clinical and Translational Research Centers (CTRC) at the University of Colorado or University of Pittsburgh. Subjects were fasted at the start of this study, and prescribed medications were withheld for the first two hours. Serial blood samples (7.5 mL) were collected at baseline and at 0.5, 1, 2, 4, 8, 12, 24, 48, 168, and 336 h into heparinized vacutainers after subjects were given a single 5000 I.U. oral dose of cholecalciferol (Jarrow Formulas, Los Angeles, CA, USA) at the start of this study. Immediately following collection, blood samples were centrifuged for 10 min at 3000× *g* at 4 °C. Plasma samples were collected and stored at −80 °C until analysis. 

### 4.2. Study Participants

Subjects diagnosed with CKD and VitD_3_ insufficiency or deficiency (25D < 30 ng/mL) and not prescribed VitD_3_ were evaluated for recruitment from the University of Colorado and University of Pittsburgh clinics (NCT02360644). For the remainder of this report, 25D_3_ <30 ng/mL was classified as VitD_3_ deficiency. All study subjects provided informed consent to participate, and the research protocols were approved by the Institutional Review Boards at the University of Colorado and the University of Pittsburgh.

### 4.3. Eligibility Criteria 

The inclusion criteria consisted of VitD_3_-deficient patients (<30 ng/mL) with hemoglobin ≥10 g/dL, age 18–75 years, likely compliance with study visits, willingness to abstain from fruit juice or alcohol within 7 days of pharmacokinetic assessments, normal hepatic function, and a diagnosis of CKD. Subjects with a predisposition to or a history of hypercalcemia, who were pregnant or lactating, who had active or recent infections requiring antimicrobial treatment, and who had autoimmune diseases with active flares were excluded from this study.

### 4.4. Analytical Assay

Ultra-high-performance liquid chromatography–tandem mass spectrometry (UHPLC-MS/MS) as previously described by Stubbs et al. [53] with minor modifications was utilized to determine the total (protein bound and unbound) plasma concentrations of VitD_3_, 25D_3_, 1,25D_3_, and 24,25D_3_. UHPLC was used for the determination of plasma concentration in this study as it was previously used in Dr. Nolin’s Study [53], and Dr. Nolin was a principal investigator with Dr. Joy on the grant. UHPLC was executed with a Waters Acquity UPLC I-class (Waters, Milford, MA, USA), which comprised a sample manager and a binary solvent manager. Concisely, acetonitrile was used to precipitate 500 µL samples followed by extraction with methyl tert-butyl ether, then derivatization with 4-phenyl-1,2,4-triazoline-3,5-dione. Derivatized VitD analytes were separated using a Waters Acquity BEH C18 column (100 mm × 2.1 mm, 1.7 µm particles) with a gradient elution of water with 0.1% formic acid and acetonitrile. The flow rate was 500 µL/min, and the total run time was 8 min. Analyte detection was achieved using positive atmospheric pressure chemical ionization and selected reaction monitoring on a TSQ Quantum Ultra triple quadrupole mass spectrometer (Thermo Scientific, San Jose, CA, USA). Standard curve ranges were 0.1–15.0 ng/mL for VitD_3_ and 24,25D_3_; 0.01–0.50 ng/mL for 1,25D_3_; and 1.0–100.0 ng/mL for 25D_3_. The mean correlation coefficients were ≥0.994 for all calibration curves. The within-run and between-run accuracy and precision percentage coefficient of variation were <10.6% for all analytes.

### 4.5. Population Pharmacokinetic Model Development 

The plasma concentrations of VitD_3_ and metabolites 25D_3_, 1,25D_3_, and 24,25D_3_ were used for nonlinear mixed effects pharmacokinetic modeling with Phoenix NLME (v.8.3, Certara Inc, Princeton, NJ, USA). Model development was performed sequentially, starting with the VitD_3_ parent compound followed by the incorporation of each subsequent metabolite. Intermediate models after the incorporation of metabolites to the parent compound were used to stabilize the model by freezing the parameters and removing random effects. The final model, including VitD_3_ and its three metabolites, was developed with simultaneous modeling (Figure 1). The observed plasma concentrations were converted to molarities from ng/mL to nmol/L in order to combine VitD_3_ (384.64 g/mol), 25D_3_ (400.64 g/mol), 1,25D_3_ (416.64 g/mol), and 24,25D_3_ (416.64 g/mol) data into a single data set. Subjects were given a single 5000 I.U. oral dose of cholecalciferol.

### 4.6. Base Model Development

Based on a visual inspection of the concentration vs. time plot and a review of the literature, one- and two-compartment structural models were evaluated for VitD_3_. Zero-order and first-order absorption with and without a lag time were explored. A noticeable feature of the VitD_3_ data was a large quantity of concentrations (>72%) below the limit of quantification (BLQ) observations. Therefore, the M1 method, which ignores BLQ values, and the M3 method, which retains all BLQ observations, were investigated [65]. Given the endogenous input of VitD_3_ through diet and sunlight sources, a zero-order endogenous production rate constant for VitD_3_ (k_endog_) was estimated as a function of VitD_3_ baseline concentration and clearance (CL/F_VitD3_). Once the final structural model for the parent compound (VitD_3_) was identified, a compartment was added for 25D_3_, the first sequential major metabolite formed. For the model to be identifiable, the fraction of VitD_3_ converted to 25D_3_ (fm_1_) was fixed to 1. The assigned value of this fraction was based on the conversion rates obtained from a 10-compartment physiologically based pharmacokinetic (PBPK) model that employed data from healthy controls who completed the same clinical study (NCT0236064) [55]. After establishing acceptable structural models for VitD_3_ and 25D_3_, two additional compartments were added to accommodate 1,25D_3_ and 24,25D_3_, respectively. The fraction of 25D_3_ converted to 1,25D_3_ (fm_2_) was fixed to 0.017 based on the previous PBPK model [55]. The remaining 25D_3_ was assumed to be converted to 24,25D_3_ through 1-fm_2_. First-order and saturable formation models were assessed for modeling metabolite concentrations. The optimal structural model was selected based on the objective function value (equal to twice the negative log likelihood [-2LL]), Akaike information criterion (AIC), and the visual inspection of goodness-of-fit (GOF) plots.

Additive, proportional, and combined additive and proportional error models were evaluated for the parent and each metabolite to explain residual variability. The inter-individual variability (IIV) in pharmacokinetic parameters assumed log-normal distributions and was evaluated according to
P_i_ = TVP exp(η_i_)
where TVP represents the population mean of the pharmacokinetic parameter, P_i_ represents the individual estimate of the pharmacokinetic parameter, and η_i_ represents the IIV.

### 4.7. Covariate Model

Several covariates were evaluated for inclusion into the model by a visual inspection of IIV versus covariate plots: weight, body mass index (BMI), age, gender, race, ethnicity, estimated glomerular filtration rate (eGFR), genetic polymorphisms in the enzymes (*CYP2R1*, *CYP27B1*, *CYP24A1*) for VitD_3_ metabolism, *VDR*, and Group-specific Component (*GC*) encoding DBP, and the protein levels of parathyroid hormone (PTH) and fibroblast growth factor 23 (FGF-23). Potential covariates identified based on visual inspection and biological plausibility were then evaluated using stepwise forward addition followed by backward elimination. Covariates at the *p* < 0.05 level were included during stepwise addition, and covariates at the *p* < 0.01 level were retained during backward elimination.

### 4.8. Model Evaluation and Validation

The selection of the final structural base model was based on the OFV, AIC, condition number, precision of fixed and random effect estimates and a visual inspection of GOF plots. IIV estimates for parameters with high h-shrinkage (>40%) were removed. The final model was validated using a visual predictive check (VPC). Data for 200 subjects were simulated using the parameter estimates from the final model. The 5th, 50th, and 95th percentiles of the predicted concentrations versus time were plotted, and observed concentrations were overlaid to evaluate the adequacy of the model.

### 4.9. Simulations

The final population pharmacokinetic model of VitD_3_ and its metabolites was used in a simulation exercise to evaluate expected 25D_3_ concentrations following common clinically prescribed cholecalciferol dosages (600 I.U./day, 1000 I.U./day, 2000 I.U./day, 5000 I.U./day, and 10,000 I.U./day) over a 6-month duration. A total of 4320 plasma concentration and associated timepoints were used, and 20 replicates were incorporated for the simulation. The mean simulated concentration was determined at each timepoint for each dose level by using the descriptive statistics on the simulation results.

## 5. Conclusions

In conclusion, we successfully developed and evaluated a comprehensive population pharmacokinetic model that adequately captures the concentration–time profiles of VitD_3_ and its three metabolites, 25D_3_, 1,25D_3_, and 24,25D_3_, in CKD subjects with VitD_3_ deficiency. Simultaneous modeling approaches may be used to explain VitD_3_ and metabolite disposition, which may be important in the CKD population. This comprehensive population pharmacokinetic model described VitD_3_ and metabolite pharmacokinetics and is an important step towards optimizing VitD_3_ dosing regimens to achieve targeted levels in the CKD population. Based on the conducted simulations, a cholecalciferol dose of 600 I.U./day to 1000 I.U./day for 6 months would be predicted to mitigate deficiency and achieve the target 25D_3_ concentration of 30–60 ng/mL. Future work will focus on evaluating regimens of cholecalciferol for VitD_3_ maintenance regimens.

## Figures and Tables

**Figure 1 ijms-25-12279-f001:**
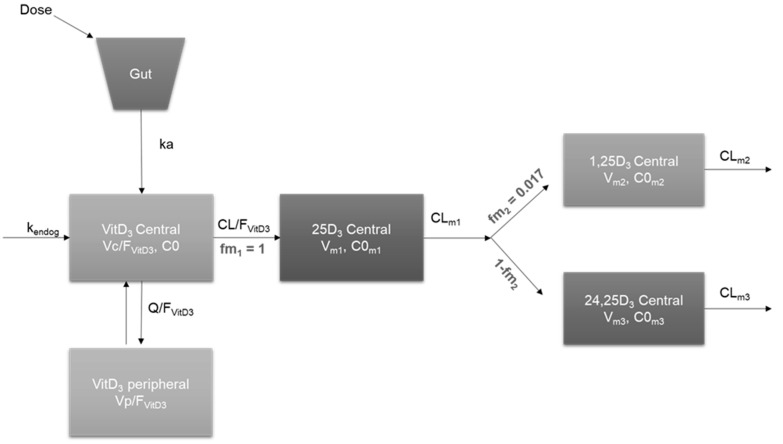
Schematic diagram of combined population pharmacokinetic model for vitamin D_3_ (VitD_3_), 25-hydroxyvitamin D_3_ (25D_3_), 1,25-dihydroxyvitamin D_3_ (1,25D_3_), and 24,25-dihydroxyvitamin D_3_ (24,25D_3_). Ka = absorption rate constant; k_endog_ = endogenous production rate constant; C0 = VitD_3_ baseline concentration; Vc/F_VitD3_ = apparent central volume of distribution of VitD_3_; CL/F_VitD3_ = apparent clearance of VitD_3_; Vp/F_VitD3_ = peripheral volume of distribution of VitD_3_; Q/F_VitD3_ = intercompartmental clearance of VitD_3_; fm1 = fraction of VitD_3_ metabolized to 25D_3_; C0_m1_ = 25D_3_ baseline concentration; V_m1_ = volume of distribution of 25D_3_; CL_m1_ = clearance of 25D_3_; f_m2_ = fraction of 25D_3_ metabolized to 1,25D_3_; C0_m2_ = 1,25D_3_ baseline concentration; V_m2_ = volume of distribution of 1,25D_3_; CL_m2_ = clearance of 1,25D_3_; C0_m3_ = 24,25D_3_ baseline concentration; V_m3_ = volume of distribution of 24,25D_3_; CL_m3_ = clearance of 24,25D_3_.

**Figure 2 ijms-25-12279-f002:**
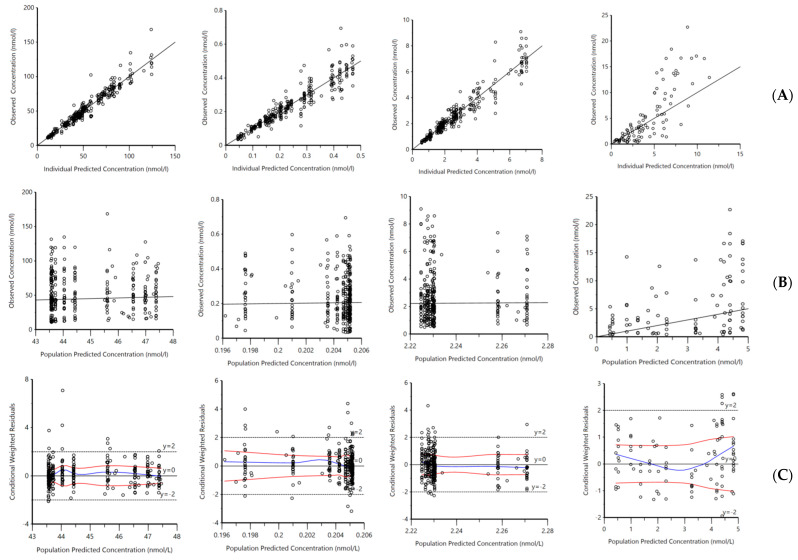
Goodness–of–fit plots, (**A**) OBS vs. IPRED, (**B**) OBS vs. PRED, (**C**) CWRES vs. PRED, and (**D**) CWRES vs. time for model-predicted (**i**) 25-hydroxyvitamin D_3_ (25D_3_) plasma concentrations, (**ii**) 1,25-dihydroxyvitamin D_3_ (1,25D_3_), (**iii**) 24,25-dihydroxyvitamin D_3_, (24,25D_3_), and (**iv**) vitamin D_3_ (VitD_3_). OBS = observed concentration; IPRED = individual predicted concentration; PRED = population predicted concentration; CWRES = conditional weighted residuals. The black solid line in (**A**,**B**) represents the line of unity. The blue solid line in CWRES plot represents trend line for linear regression and red solid line is used to observe the distribution trend of residuals.

**Figure 3 ijms-25-12279-f003:**
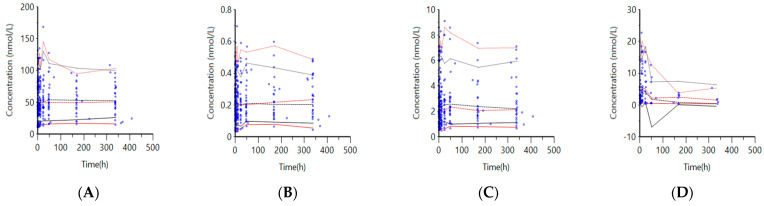
VVPCs for the final model. Observed concentrations (circle) and 5th (solid line), 50th (dashed line), and 95th (dotted line) percentiles from observed (red) and predicted (blue) data for (**A**) 25-hydroxyvitamin D_3_ (25D_3_), (**B**) 1,25-dihydroxyvitamin D_3_ (1,25D_3_), (**C**) 24,25-dihydroxyvitamin D_3_ (24,25D_3_), and (**D**) vitamin D_3_ (VitD_3_).

**Figure 4 ijms-25-12279-f004:**
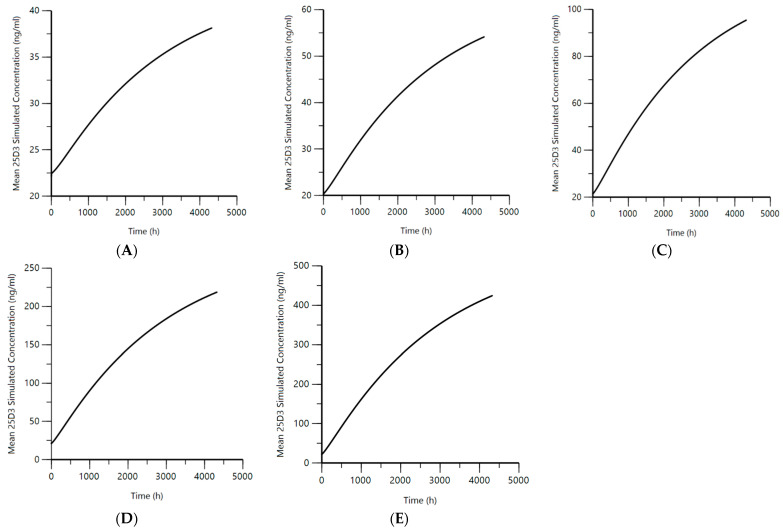
Population simulations for mean 25D_3_ concentrations based on daily VitD_3_ doses over six-month time period. (**A**) mean 25D_3_ simulated concentration vs. time for dose of 600 I.U./day, (**B**) mean 25D_3_ simulated concentration vs. time for dose of 1000 I.U./day, (**C**) mean 25D_3_ simulated concentration vs. time for dose of 2000 I.U./day, (**D**) mean 25D_3_ simulated concentration vs. time for dose of 5000 I.U./day, (**E**) mean 25D_3_ simulated concentration vs. time for dose of 5000 I.U./day.

**Table 1 ijms-25-12279-t001:** Baseline characteristics of study participants (*n* = 29).

Gender		
Female	17 (59%) 12 (41%)	
Male		
Race		
White	19 (66%) 10 (34%)	
Black		
Ethnicity ^a^		
Non-Hispanic	24 (83%) 4 (14%)	
Hispanic		
Age (years)	61 (29–73)	
Weight (kg)	92.0 (70.7–135.3)	
BMI (kg/m^2^)	32.6 (25.6–43.4)	
eGFR (mL/min/1.73m^2^)	37 (11–97)	
Stage 1	1 (3%)	
Stage 2	5 (17%)	
Stage 3	14 (48%)	
Stage 4	8 (28%)	
Stage 5	1 (3%)	
25D_3_ (ng/mL)	18 (7–29)	
Total no of VitD3 samples	310	
No of BLQ VitD3 samples	212	
*CYP27B1* rs10877012		
C/C	15 (51%)	
C/A	11 (38%)	
*CYP27B1* rs10877012		
A/A		2 (7%)
ND		1 (3%)
*CYP2R1* rs12794714		
G/G		15 (51%)
G/A		13 (45%)
A/A		0
ND		1 (3%)
*CYP24A1* rs6013897		
A/A		18 (62%)
A/S		8 (28%)
S/S		2 (7%)
ND		1 (3%)
*GC*_VDBP rs7041		
G/G		17 (59%)
G/A		9 (31%)
A/A		2 (7%)
ND		1 (3%)
*VDR* rs2228570		
G/G		17 (59%)
A/G		6 (21%)
A/A		5 (7%)
ND		1 (3%)
VDR rs7968585		
G/G		7 (24%)
G/A		14 (49%)
A/A		7 (24%)
ND		1 (3%)

Data are presented as median (range) or number (%). Abbreviations: eGFR—estimated glomerular filtration rate; 25D_3_—calcidiol, ND—not determined. ^a^ One subject declined to disclose their ethnicity.

**Table 2 ijms-25-12279-t002:** Population pharmacokinetic parameters of VitD3 and major metabolites.

Parameter	Symbol	Estimate (CV%)
VitD_3_ baseline concentration (nmol/L)	C0	0.98 (41.7)
Absorption rate constant (h^−1^)	ka	Fixed to 0.054
Endogenous production rate constant (nmol/h)	k_endog_	Fixed to 0.55
VitD_3_, apparent central volume of distribution (L)	V_C_/F_VitD3_	21.3 (22.2)
VitD_3_, apparent clearance (L/h)	CL/F_VitD3_	1.4 (42.4)
VitD_3_, peripheral volume of distribution (L)	Vp/F_VitD3_	Fixed to 50
VitD_3_, intercompartmental clearance (L/h)	Q/F_VitD3_	Fixed to 0.44
25D_3_, baseline concentration, (nmol/L)	C0_m1_	43.5 (4.1)
25D_3_, volume of distribution (L)	V_m1_	58.3 (14.8)
25D_3_, clearance (L/h)	CL_m1_	0.02 (52.2)
1,25D_3_, baseline concentration (nmol/L)	C0_m2_	0.20 (6.9)
1,25D_3_, volume of distribution (L)	V_m2_	71.5 (206.8)
1,25D_3_, clearance (L/h)	CL_m2_	0.08 (47.7)
24,25D_3_, baseline concentration (nmol/L)	C0_m3_	2.2 (9.4)
24,25D_3_, volume of distribution (L)	V_m3_	105.2 (140.5)
24,25D_3_, clearance (L/h)	CL_m3_	0.40 (53.4)
Residual error, VitD_3_	σ_1_	12.5 (3.1)
Residual error, 25D_3_	σ_2_	65.7 (23)
Residual error, 1,25D_3_	σ_3_	17.2 (4.6)
Residual error, 24,25D_3_	σ_4_	16.6 (5.7)

CV = coefficient of variation.

## Data Availability

The data that support the results of this study are available from the corresponding author upon reasonable request.

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
