# Peer review of "Population Pharmacokinetic Model of Vitamin D3 and Metabolites in Chronic Kidney Disease Patients with Vitamin D Insufficiency and Deficiency"

_ijms, 2024, doi:10.3390/ijms252212279_

Round 1

Reviewer 1 Report

Comments and Suggestions for Authors

Dear Editor,

Thank you for inviting me to review this article. I read the manuscript with interest and appreciate the detailed exploration of the pharmacokinetic model of Vitamin D3 in patients with chronic kidney disease, a highly relevant and timely topic. The article has several strengths, including a detailed methodology and a well-structured pharmacokinetic model, with potential clinical application in personalizing Vitamin D3 doses for CKD patients. However, I have identified some areas that could be improved to enhance the study’s accessibility and robustness.

Below is a list of more specific suggestions and revisions organized by line numbers.

Specific Line-by-Line Modifications

  1. Lines 21-33 (Abstract)
    Add a concluding sentence to connect the results with future clinical applications.
    Text suggestion: “These findings may contribute to developing personalized dosing strategies for Vitamin D3 treatment in the CKD population.”

  2. Line 56
    Update the citation to support the prevalence of Vitamin D insufficiency in CKD patients. I suggest including more recent studies.

  3. Lines 69-70
    Rewrite to clarify the relationship between CKD and the variability of Vitamin D3 dosing.
    Text suggestion: “The CKD population exhibits significant variability in kidney function, body composition, and comorbidities, factors that complicate dose-response relationships for Vitamin D3.”

  4. Methods Section
    It would be advisable to start with a section titled “Study Design,” describing the type of study, followed by the “Setting and Participants” section. Also, include a section detailing the inclusion criteria and sample selection.

  5. Line 85
    Specify the clinical relevance of the parameters listed (e.g., age, eGFR) to help the reader better understand variability among participants. Add a note highlighting how these parameters may impact the results.

  6. Lines 163-168 (Figure 2)
    Figure 2 is difficult to interpret due to its small size. Could it be adjusted to improve clarity?

  7. Lines 189-206 (Section 2.4)
    Briefly elaborate on the choice of simulated doses, emphasizing their alignment with current clinical practices.
    Text suggestion: “The simulated doses (600-10,000 I.U.) reflect clinically relevant dosages commonly used to treat Vitamin D insufficiency in CKD patients.”

  8. Lines 254-256
    Specify why the absorption parameter (ka) was set to 0.054 h-1, as this may appear arbitrary.
    Text suggestion: “The ka value was fixed based on estimates derived from previous iterations of the model, as there was insufficient data in the absorption phase to calculate it.”

  9. Line 339
    Suggested revision: Include a sentence explaining that, although no covariate reached statistical significance, some may have practical implications on pharmacokinetic parameters.
    Text suggestion: “Despite the lack of statistical significance, certain covariates, such as weight and BMI, could have practical relevance for Vitamin D3 pharmacokinetics.”

  10. Line 479 (Conclusions)
    Briefly summarize how the pharmacokinetic model contributes to treatment personalization.

  11. References
    The bibliography includes some older studies; if possible, consider replacing these with more recent studies.

Author Response

#REVIEWER 1:

Thank you for inviting me to review this article. I read the manuscript with interest and appreciate the detailed exploration of the pharmacokinetic model of Vitamin D3 in patients with chronic kidney disease, a highly relevant and timely topic. The article has several strengths, including a detailed methodology and a well-structured pharmacokinetic model, with potential clinical application in personalizing Vitamin D3 doses for CKD patients. However, I have identified some areas that could be improved to enhance the study’s accessibility and robustness.

Below is a list of more specific suggestions and revisions organized by line numbers.

Specific Line-by-Line Modifications

  1. Lines 21-33 (Abstract) Add a concluding sentence to connect the results with future clinical applications.

Text suggestion: “These findings may contribute to developing personalized dosing strategies for Vitamin D3 treatment in the CKD population.

Thank you for the suggestion. The concluding sentence has been added to the abstract as: These simulation findings could potentially contribute to the development of personalized dosage regimens for Vitamin D treatment in patients with CKD.

  1. Line 56 Update the citation to support the prevalence of Vitamin D insufficiency in CKD patients. I suggest including more recent studies.

Citation has been updated with more recent studies as :  Patients with chronic kidney disease (CKD) are among the most vulnerable populations at risk for VitD3 deficiency with prevalence rates of up to 80% previously reported.14,15-17

  1. Chiriac C et al. Jan-Mar 2024;20(1):12-20. doi:10.4183/aeb.2024.12
  2. Lee J et al. 2023;10:1017459. doi:10.3389/fmed.2023.101745
  3. Franca G Ph et al. Aug 17 2018;15(8)doi:10.3390/ijerph15081773

  1. Lines 69-70 Rewrite to clarify the relationship between CKD and the variability of Vitamin D3 dosing. Text suggestion: “The CKD population exhibits significant variability in kidney function, body composition, and comorbidities, factors that complicate dose-response relationships for Vitamin D3.”

Thank you for the suggestion. The text is now rewritten as: The CKD population exhibits substantial variation in renal function, body composition, comorbidities, and concomitant medications that complicate dose-response relationships for VitD3.

  1. Methods Section It would be advisable to start with a section titled “Study Design,” describing the type of study, followed by the “Setting and Participants” section. Also, include a section detailing the inclusion criteria and sample selection

The method section now begins with Study Design, followed by Study Participants and Inclusion Criteria as recommended.

  1. Line 85 Specify the clinical relevance of the parameters listed (e.g., age, eGFR) to help the reader better understand variability among participants. Add a note highlighting how these parameters may impact the results.

To specify the clinical relevance and help the reader understand the impact of the parameters listed in the results, a few lines have been added.

“Several targeted patient parameters were assessed given their potential impact on VitD3 metabolism and concentrations. Weight and BMI can have an inverse relationship to VitD3 concentrations, as tissue distribution increases with increased body fat. Age related physiological changes can impact the metabolism of VitD3 and eGFR, a marker of renal function, can impact the catabolism of VitD3.

  1. Lines 163-168 (Figure 2) Figure 2 is difficult to interpret due to its small size. Could it be adjusted to improve clarity?

The quality of Figure 2 has been enhanced compared to the previous version, however due to the limitation of size within the manuscript, it was not enlarged.

  1. Lines 189-206 (Section 2.4) Briefly elaborate on the choice of simulated doses, emphasizing their alignment with current clinical practices. Text suggestion: “The simulated doses (600-10,000 I.U.) reflect clinically relevant dosages commonly used to treat Vitamin D insufficiency in CKD patients.

The simulations were conducted based on cholecalciferol dosage regimens used in the clinic and represented a range of low, middle and high doses."

  1. Lines 254-256 Specify why the absorption parameter (ka) was set to 0.054 h 1, as this may appear arbitrary. Text suggestion: “The ka value was fixed based on estimates derived from previous iterations of the model, as there was insufficient data in the absorption phase to calculate it.”

Thank you for recommending an addition of justification on fixing Ka. The specification for fixing Ka value to 0.054 1/h is added as: The ka was fixed to 0.054 h-1 based on estimations derived from previous iterations of the model as there was inadequate data in the absorption phase for its estimation.

  1. Line 339 Suggested revision: Include a sentence explaining that, although no covariate reached statistical significance, some may have practical implications on pharmacokinetic parameters. Text suggestion: “Despite the lack of statistical significance, certain covariates, such as weight and BMI, could have practical relevance for Vitamin D3 pharmacokinetics.”

Updated text is: While none of the covariates assessed in the current study significantly affected parameter variability, it is plausible that this study lacked enough statistical power to detect significant covariates and therefore we cannot rule out their influence on disposition of VitD3 and its metabolites. Certain covariates such as weight and BMI could have clinical relevance on VitD3 pharmacokinetics.

  1. Line 479 (Conclusions) Briefly summarize how the pharmacokinetic model contributes to treatment personalization.

A Summary statement has been added to conclusion as: This comprehensive population pharmacokinetic model described VitD3 and metabolite pharmacokinetics and is an important step towards optimizing VitD3 dosing regimens to achieve targeted levels in the CKD population. Based on the conducted simulations, a cholecalciferol dose of 600 I.U./day to 1000 I.U./day for 6 months would be predicted to mitigate deficiency and achieve target 25D3 concentration of 30-60 ng/mL. Future work will focus on evaluating regimens of cholecalciferol for VitD3 maintenance regimens.

  1. References The bibliography includes some older studies; if possible, consider replacing these with more recent studies.

Some recent studies have been added wherever feasible (REF 15-17).

Reviewer 2 Report

Comments and Suggestions for Authors

This great study on vitamin-D levels (and metabolites) in CKD-patients. Although small in size, both the set-up and work-up are very well done, along with strong statistics. I have some comments:

Methods: several analyzing methods for vitamin D exist, so you need to explain your choice for UHPLC.

Results: Table 1: did you look at a difference between male-female and/or caucasian-black study subjects?

Discussion: only calcitriol is effective in elevating vitamin D-levels in CKD-patients, which should be discussed here.

Author Response

# REVIEWER 2

This great study on vitamin D levels (and metabolites) in CKD patients. Although small in size, both the set-up and work-up are very well done, along with strong statistics. I have some comments:

  1. Methods: several analyzing methods for vitamin D exist, so you need to explain your choice for UHPLC.

The referenced UHPLC assay (ref 53) was previously developed in Dr. Nolin’s laboratory.   Dr. Nolin was a Principal investigator with Dr. Joy on the grant.

  1. Results: Table 1: did you look at a difference between male-female and/or caucasian-black study subjects?

We assessed patient gender and race in the initial visual inspection of covariate plots for each clearance and volume term in the model.  None of these covariate plots showed patterns suggestive of an impact on the clearance or volume of VitD3 or its metabolites.

  1. Discussion: only calcitriol is effective in elevating vitamin D-levels in CKD-patients, which should be discussed here.

Calcitriol is the 1,25-VitD3 metabolite.  In end-stage kidney disease patients, one must administer calcitriol directly to impact 1,25-VitD3 concentrations.  The patients in the current study had chronic kidney disease (lowest eGFR 30 ml/min), but were not Stage 4-5 CKD and none were end-stage requiring dialysis.  As our goal was not to directly impact 1,25-VitD3 concentrations, but to impact 25-VitD3 concentrations given the current well-established clinical practice and wealth of literature supporting the benefits of adequate 25-VitD3 in patients with CKD, we assessed cholecalciferol which requires one metabolic step to produce the 25-VitD3 metabolite. 

Round 2

Reviewer 1 Report

Comments and Suggestions for Authors

The authors have made the necessary revisions to the manuscript. It is considered ready for publication.